# *Juncus Bulbosus* Tissue Nutrient Concentrations and Stoichiometry in Oligotrophic Ecosystems: Variability with Seasons, Growth Forms, Organs and Habitats

**DOI:** 10.3390/plants10030441

**Published:** 2021-02-26

**Authors:** Therese F. Moe, Dag O. Hessen, Benoît O. L. Demars

**Affiliations:** 1Norwegian Institute for Water Research (NIVA), Gaustadalléen 21, 0349 Oslo, Norway; therese.fosholt.moe@niva.no; 2Department of Biosciences, University of Oslo, P.O. Box 1066 Blindern, 0316 Oslo, Norway; d.o.hessen@mn.uio.no

**Keywords:** plant stoichiometry, carbon, nitrogen, phosphorus, river, lake, nutrient

## Abstract

Aquatic plant nutrient concentrations provide important information to characterise their role in nutrient retention and turnover in aquatic ecosystems. While large standing biomass of aquatic plants is typically found in nutrient-rich localities, it may also occur in oligotrophic ecosystems. *Juncus bulbosus* is able to form massive stands even in very nutrient-dilute waters. Here we show that this may be achieved by tissues with very high carbon-to-nutrient ratios combined with perennial (slow) growth and a poor food source for grazers inferred from plant stoichiometry and tissue nutrient thresholds. We also show that the C, N, P and C:N:P stoichiometric ratios of *Juncus bulbosus* vary with the time of year, habitats (lakes *versus* rivers) and organs (roots *versus* shoots). We found no differences between growth forms (notably in P, inferred as the most limiting nutrient) corresponding to small and large plant stands. The mass development of *J. bulbosus* requires C, N and P, whatever the ecosystem (lake or river), and not just CO_2_ and NH_4_, as suggested in previous studies. Since macrophytes inhabiting oligotrophic aquatic ecosystems are dominated by isoetids (perennial plants with a high root/shoot ratio), attention should be paid to quantifying the role of roots in aquatic plant stoichiometry, nutrient turnover and nutrient retention.

## 1. Introduction

Aquatic plants tend to have higher N and P and lower C:N and C:P ratios than terrestrial plants [1,2]. This generally corresponds to higher growth rates and decomposition rates, as well as higher herbivory [1,2,3]. Standing biomasses of freshwater macrophyte meadows are equivalent to those of grassland ecosystems [1]; hence, aquatic plants can actively contribute to nutrient cycling [4,5,6], as well as through the uptake, retention and release of nutrients from the sediment to the water column via decomposition and herbivory [7,8].

The role of aquatic plants in nutrient net retention may be very modest relative to external loading in nutrient-rich ecosystems. For example, studies on British calcareous lowland rivers impacted by agriculture and point-source effluents, bearing high standing biomass [9], reported very small plant nutrient retention relative to total flux: less than 1 to 2.5% for P [10,11,12] and 0.2 to 2% for N [10,11]. Slightly higher retention rates by aquatic plants (up to 10–13% of the dissolved inorganic N river flux during the summer) were reported in a lowland river in the Netherlands [13]. Large standing biomass of aquatic plants is also known from oligotrophic aquatic ecosystems [14,15,16], where aquatic plant nutrient retention may be more significant [7,16].

Aquatic plant nutrient retention is often determined for the shoot without the roots, but in nutrient-poor aquatic ecosystems, aquatic plants tend to be dominated by isoetids characterised by high root/shoot ratios [17]. There were more differences between roots and shoots for N than P concentrations in *Lobelia dortmanna*, especially during the summer, with larger N concentrations in shoots than roots [18]. Relative differences in N and P between roots and shoots can be driven by light and nutrient availability [19,20,21]. In north-west Europe, few aquatic plant species grow to form large submerged stands (1–3 m tall) in oligotrophic ecosystems, and among those species, *Juncus bulbosus* is the only (facultative) perennial type, also playing an important role in ecosystem structure and functions [22]. Since *J. bulbosus* is also widely distributed in northern Europe [23], it makes it an important species to study.

In southern Norway, the mass development of *Juncus bulbosus* occurs in lakes and rivers (with up to 500–1100 g dry mass m^−2^ [24,25]), despite extremely low nutrient concentrations [26,27]. *J. bulbosus* can have a high root/shoot ratio in flowing water (1.5 ± 0.5, based on dry mass m^−2^, Demars, unpublished), and so root nutrient retention should not be overlooked [28]. A detailed study on Norwegian lakes identified several local- and catchment-scale drivers of *J. bulbosus* tissue stoichiometry [26]. Some patterns in the tissue C, N, P concentrations and C:N:P stoichiometry of *J. bulbosus* in Norwegian lakes and rivers were identified [27] using averaged plant data from different years and analysed in different ways (entire plant *versus* shoot). Thus, it remains to directly test for more specific sources of variability such as seasons, growth forms, organs and habitats.

We aimed to characterise *Juncus bulbosus* tissue nutrient concentrations and stoichiometry in southern Norway, an important step towards characterising the potential role of macrophytes in nutrient retention in oligotrophic aquatic ecosystems. Here we set out to (i) test whether *Juncus bulbosus* C, N, P and C:N:P in lakes can be related to growth forms (abundance) and the time of year; (ii) test for differences in *Juncus bulbosus* elemental concentrations and stoichiometry between habitats (lakes *versus* rivers), plant organs (roots *versus* shoots) and habitats × plant organs; and (iii) assess the likelihood of nutrient limitation for yield (standing biomass) using a comparative approach [26].

## 2. Results

### 2.1. Growth Forms and the Time of Year

C, N, P (46 ± 2, 1.7 ± 0.2, 0.16 ± 0.08%) and C:N:P (1073:31:1) in *Juncus bulbosus* varied greatly and significantly between the six lakes surveyed in 2006. There were, however, no significant differences in C, N, P and C:N:P between large growth forms and rosette leaves or new shoots. The effect of growth form on N concentrations in plant tissue (close to significance after correction for multiple testing) explained 9.3% of the variance, whereas the full mixed-effects model (including random effects) explained 34% of the variance. This was mostly due to differences (median) in N tissue concentrations in the large growth form (June: 2.2%, October: 2.6%) *versus* rosette (June: 2.6%, October: 2.8%). The effect of the time of sampling (June *versus* October) on P concentrations in plant tissue explained 14% of the variance, and the full model explained 49% of the variance. The median P concentration was lower in October (0.09%) than in June (0.13%), and correspondingly, the C:P and N:P were higher: C:P = 1332 *versus* 954; N:P = 61 *versus* 45, respectively (Table 1).

C, N, P (46 ± 2, 1.7 ± 0.3, 0.18 ± 0.09%) and C:N:P (843:26:1) in *Juncus bulbosus* were also very variable in the ten lakes surveyed in summer 2008. Neither growth form nor the time of sampling (June *versus* September) could explain the variability in C, N, P or C:N:P (Table 2).

### 2.2. Plant Organs (Roots versus Shoots) and Habitats (Lakes versus Rivers)

We first checked for differences in environmental conditions. The 16 lakes and 28 river sites sampled for this purpose differed in this context. The lakes had higher total N concentrations (median: 300 *versus* 230 µg N L^−1^, F = 19, *P* = 4 × 10^−5^), a higher sediment organic matter content (18 *versus* 2%, F = 22, *P* = 1 × 10^−5^), a higher proportion of sediment pore water (76 *versus* 36%, F = 28, *P* = 1×10^−6^) and higher electric conductivity (32 *versus* 16 µS cm^−1^, F = 41, *P* = 1 × 10^−10^). There were no differences in inorganic N or P concentrations either from the sediment or the water column between lakes and rivers.

C tissue concentration was similar across organs and habitats (Figure 1). The roots had a lower N concentration (F = 86, *P*= 4 × 10^−14^) and lower P concentration (F = 13, *P* = 0.0006) than the shoots. N tissue concentration was higher in the rivers than the lakes (F = 20, *P* = 2 × 10^−5^) and similarly for P, although not statistically significant. C:N and C:P reflected N and P, but N:P was similar across organs and habitats. There were no significant interactions between organs and habitats. The average shoot concentrations were 1.76 *versus* 2.40% for N and 0.12 *versus* 0.16% for P in lakes *versus* rivers, respectively. The average shoot C:N:P ratios were 1091:35:1 *versus* 824:38:1 in lakes *versus* rivers, respectively. Multivariate analyses identified organs (shoot *versus* root) as the main factor able to explain the variance in plant tissue concentrations and stoichiometry, together with habitats (lake *versus* river), total P and electric conductivity, independently of the way missing data were handled (Table 3). The forward selection of these significant factors produced a highly significant (*P* < 0.001) model with an adjR^2^ of 30% (Table 3).

## 3. Discussion

Overall, there were little or no differences in plant element concentrations and stoichiometry between growth forms (2006 and 2008 surveys), similarly to between small and large plant stands [27], likely because the plant growth rate was nutrient limited, and the N:P molar ratio was relatively strict [26]. The possible difference in N concentrations (2006 survey only) between growth forms may be due to unbalanced P limitation leading to N “luxury consumption” (Figure 2a).

There were no temporal differences in C, N, P and C:N:P between samples collected in early *versus* late summer (2008 survey) but large differences in P, C:P and N:P ratios between samples collected in late spring *versus* autumn (2006 survey). Such seasonal shifts in plant stoichiometry have been reported before for other species and are known to vary between species and elements [29,30,31,32]. In *Lobelia dortmanna* (an isoetid), N accumulation occurred under low light availability and slow growth in the autumn for possible use in the following spring and summer growth spurt [18].

The most imbalanced stoichiometry was from samples collected in autumn when the plants had much higher N relative to P, likely indicative of a period with slow growth, indicated by the nutrient thresholds derived from laboratory bioassays (Figure 2a [18,26]). *Juncus bulbosus* had a lower N tissue concentration in lakes than rivers in southern Norway (2010 survey) despite similar external inorganic nutrient concentrations (water and sediment). This may reflect the higher supply rate of nutrients in rivers compared to lakes, especially the more limiting factor P, with most samples below the critical N:P ratio (Figure 2b). Phosphorus limitation may also be more pronounced in lakes than rivers due to a higher organic matter content in lake than river sediments (18% *versus* 2%, respectively), potentially triggering root iron plaque formation and preventing P uptake, as for other isoetids [33]. The difference between habitats (standing *versus* flowing water) was much more pronounced than in a previous study, where the effects were largely masked by plant life forms and biophysical zones [34]. This said species evolved in different environments, and their response to contrasting habitats such as current velocity [29] or lotic *versus* lentic habitats [28] is known to differ.

The higher nutrient concentration in shoots compared to roots can be explained by the high N concentrations abundant in chlorophyll and Rubisco (the enzyme-fixing CO_2_) in the leaf, as well as the energetic requirements associated with photosynthesis, requiring a high concentration of the P-rich adenosine 5′-triphosphate (ATP) [35].

The average P concentrations in the shoots of *Juncus bulbosus* (0.12% in lakes and 0.16% in rivers) in summer 2010 in Norway were generally lower than *Juncus bulbosus* and associated species recorded in Scotland (*Juncus bulbosus* P = 0.22% and molar C:P = 639—Figure 3 [14]) but comparable to some species in Spanish oligotrophic lakes (e.g., *Potamogeton natans* P = 0.12% and molar C:P = 965; *Littorella uniflora* P = 0.12% and C:P = 861; *Juncus heterophyllus* P = 0.09% and C:P = 1218—Figure 3 [36]). These average concentrations suggested no limitations for maximum yield (critical threshold: P = 0.14%, Figure 3) but limitations for maximum growth rates (critical threshold: P = 0.22% [26]). These low P concentrations and high C:P molar ratios (lake C:P = 1091 and rivers C:P = 824) in plant tissue combined with perennial growth give the opportunity to form mass stands under extremely low nutrient regimes. They also make the plant relatively unattractive to generalist grazers [2,8] in addition to being a cyanogenic plant [37]. In southern Norway, aquatic plants including *Juncus bulbosus* were also found to have higher heavy metal concentrations in their tissue than terrestrial plants but suggested aquatic plants as a source of sodium for moose [38].

Reading through the original publications [24,39,40], no factorial experiments were ever carried out to test the independent and interactive effects of CO_2_, N and P on the *Juncus bulbosus* growth rate and yield. The liming of lakes in Norway promoted *J. bulbosus* mass development through the mineralisation of the organic matter with the production of CO_2_, NH_4_ and inorganic P in sediment pore water [24] until the organic matter became less reactive over time [41]. In the presence of sufficient N and P, CO_2_ can be a limiting factor [39,40,42]. Excess of CO_2_ availability relative to N and P is expected to increase the C:N and C:P of autotrophs in nutrient-poor freshwater ecosystems [43,44,45]. General N and/or P limitations have been reported for the phytoplankton of Norwegian lakes [46,47] and inferred through *J. bulbosus* plant tissue nutrient concentration and stoichiometry [26].

The repeated story that the mass development of *J. bulbosus* would only occur under high CO_2_ and sediment pore water NH_4_ concentrations [24,27,41,48] can only make sense if the plant can access enough P for its growth because aquatic plant tissue C:N:P stoichiometry is rather homeostatic [14,26], whatever the ecosystem, based on principles of ecological stoichiometry. Thus, in Norwegian lakes and rivers, it is likely that CO_2_, N and/or P could limit or colimit the growth rate of *J. bulbosus*. Certainly, the role of P should not be disregarded (this study [26]). *J. bulbosus* has also been able to produce stands with large biomass in (unlimed) oligotrophic ecosystems through perennial growth where physical conditions allow (high light, low water turbulence) (Figure 4 [49]). There were no significant differences in the water column and sediment pore water nutrient concentrations between stands with low and high *J. bulbosus* biomass in rivers [50], unlike for lakes shortly after liming [24].

## 4. Materials and Methods

### 4.1. Juncus Bulbosus

*Juncus bulbosus* is a perennial species common in oligotrophic and ultraoligotrophic European lakes and rivers [51,52]. It has caused problems with mass development along the littoral zones of many lakes and rivers since the mid-1980s [53,54,55]. Although these problems have receded in some areas [41,56], it remains an issue in southern Norway [15,27] (Figure 4). *J. bulbosus* is an amphibious plant morphologically very plastic, ranging from small tufted terrestrial plants to submerged, floating aquatics, often rooting at the nodes and freely fruiting [57]. In this study, all samples were rooted underwater, and thus *J. bulbosus* may be referred to as f. *aquatica*, with the largest shoots reaching the water surface.

### 4.2. Growth Form and the Time of Year

In a preliminary study, 86 plant samples (3–10 shoots per lake) were separated into rosette leaves, large growth forms (stem or “column” with annual shoots, as illustrated in [15] and Figure 4) and new annual shoots (rosette-bearing segments) representing the vegetative life cycle of *J. bulbosus*, collected in southern Norway from six lakes in spring (31 May–14 June) and autumn (29 September–12 October, 2006) (Figure 4) [58].

We collected an additional set of plant samples (*n* = 40) from ten lakes with two growth forms (rosette and large growth forms, both present in the ten lakes) and two summer sampling times (21–26 June 2008 and 6–10 September 2008) per lake. The data are available in the Appendix A.

### 4.3. Plant Organs (Roots versus Shoots) and Habitats (Lakes versus Rivers)

Sixteen lakes were sampled during 26–30 July 2010 and 28 river sites (50 m stretches from 15 different rivers) during 4–16 August 2010 to analyse the roots and shoots of *J. bulbosus.* Sites were chosen to span a wide range of *J. bulbosus* growth forms and abundances. A single individual (root and shoot) of *J. bulbosus* per site was picked from the dominant type of growth form or most abundant stand, as previously described [26,27]. Thus, rosette growth forms generally corresponded to stands with low biomass, whereas large growth forms corresponded to stands with large biomass.

The concentrations of NH_4_, NO_3_, PO_4_, Ca, dissolved inorganic carbon (DIC), CO_2_, total organic carbon (TOC), total N and total P, as well as pH and electric conductivity, were determined for the water column. The concentrations of NH_4_, NO_3_ and PO_4_ in sediment pore water, as well as the percentage of organic matter and proportion of pore water in sediments, were also determined. All samples were collected and analysed, as described in previous studies [26,27]. The data are available in the Appendix A.

### 4.4. Plant Tissue C, N, P Analyses

All plants were cleaned free of detritus and periphyton by hand. The plants were then freeze dried in 2006 [58] or air dried at room temperature for the 2008 and 2010 datasets [27]. The dried plants were ground prior to analyses for C, N and P, as previously described [26,27].

### 4.5. Statistics

In total, we used three sets of data from southern Norway, briefly summarised in Table 4. Two of the datasets (2008, 2010) were partially used in a previous study designed to test different hypotheses, as explained in the Introduction (Section 1) [27]. Here we did not combine those datasets as in [27] as they were not comparable, and we did not use plant nutrient concentrations and stoichiometry as factors to explain plant abundance (corresponding to growth form) as in [27] but as response variables to test for the effects of seasons, growth forms, organs and habitats. We then interpreted those datasets differently to [27] with a comparative approach based on a review of laboratory bioassays, as previously applied to a dataset collected in 2007 [26].

From the lakes surveyed in 2006 and 2008, we tested for differences in C, N, P and C:N:P with mixed-effects models (allowing for unbalanced design) using sampling time or growth form as the fixed effects and site as the random effect. Sampling time or growth form were also used as random effects when testing for growth form or sampling time, respectively. The response variables were log transformed prior to analyses to normalise the data. Differences were considered significant after correcting for multiple testing using a Bonferroni correction (α = 0.05/n, with n number of tests). Statistics were computed in R version 3.5.0 [59] using the lme4 package [60]. The fit of the models with significant fixed effects was assessed with the conditional and marginal coefficient of determination (R^2^) using the R function r.squaredGLMM from the MuMIn package [61,62]. The conditional R^2^ represents the variance explained by fixed and random factors together, and the marginal R^2^ represents the variance explained solely by the fixed effects.

From the lakes and rivers surveyed in 2010, we first tested whether there were any statistical differences in environmental variables, notably in nutrient concentrations (water and sediment) using one-way ANOVA in R. We then tested for the effect of habitats (lake and river), organs (root and shoot) and their interaction on C, N, P and stoichiometric molar ratios using two-way ANOVA in R version 3.5.0 [59]. It was not necessary to apply any data transformations prior to statistical analyses in this dataset.

In addition, we used multivariate analyses (redundancy analyses) to test for the single and collective effects of organs (root, shoot), habitats (lakes, rivers) and environmental variables (water and sediment chemistry) on plant nutrient concentrations (C, N, P) and stoichiometry (C:N, C:P, N:P) using the 2010 dataset. We also calculated sediment pore water nutrients per volume of sediment (concentrations × pore water volume). Missing values for plant P tissue in 10 samples and sediment pore water in 11 samples (out of 80 samples) were handled in two different ways, either by replacing missing data with averages taken across all samples or simply removing the samples with missing data. This led to virtually the same results. Electric conductivity, NH_4_ and sediment pore water concentrations were ln transformed to normalise the data. The significance of individual predictors was tested by running 999 unrestricted Monte Carlo random permutations. We then use a forward stepwise selection of significant variables to produce the best models. The multivariate analyses were run with Canoco 5.0 (Microcomputer Power: Ithaca, NY, USA) [63].

## 5. Conclusions

There was no (or very little) difference in plant nutrient concentrations between growth forms but large differences between the time of year, habitats and organs. The very high C:P ratios recorded here suggested *Juncus bulbosus* is not attractive to grazers and combined with (slow) perennial growth can reach high standing biomass in nutrient-poor lakes and rivers as observed. In order to quantify nutrient retention of aquatic plants in oligotrophic aquatic ecosystems, it will be important to sample year-round and quantify the contribution of roots and shoots.

## Figures and Tables

**Figure 1 plants-10-00441-f001:**
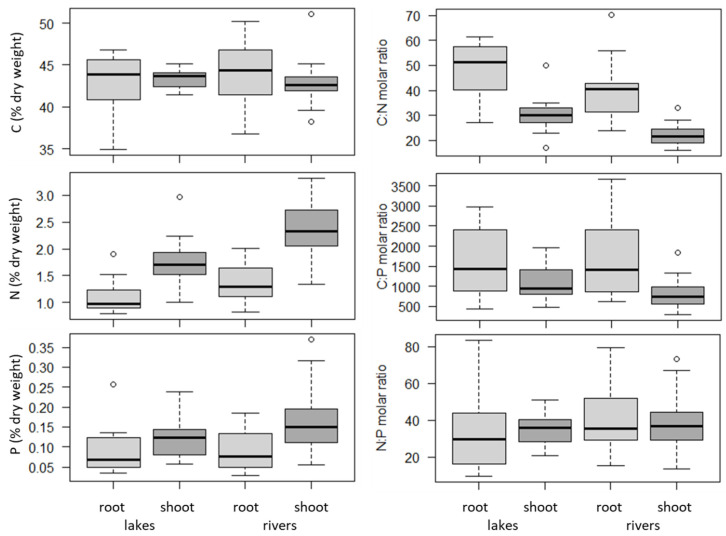
Differences in the element concentrations and stoichiometry of *Juncus bulbosus* between organs (roots and shoots) and habitats (lakes and rivers)—summer 2010 (Table 4).

**Figure 2 plants-10-00441-f002:**
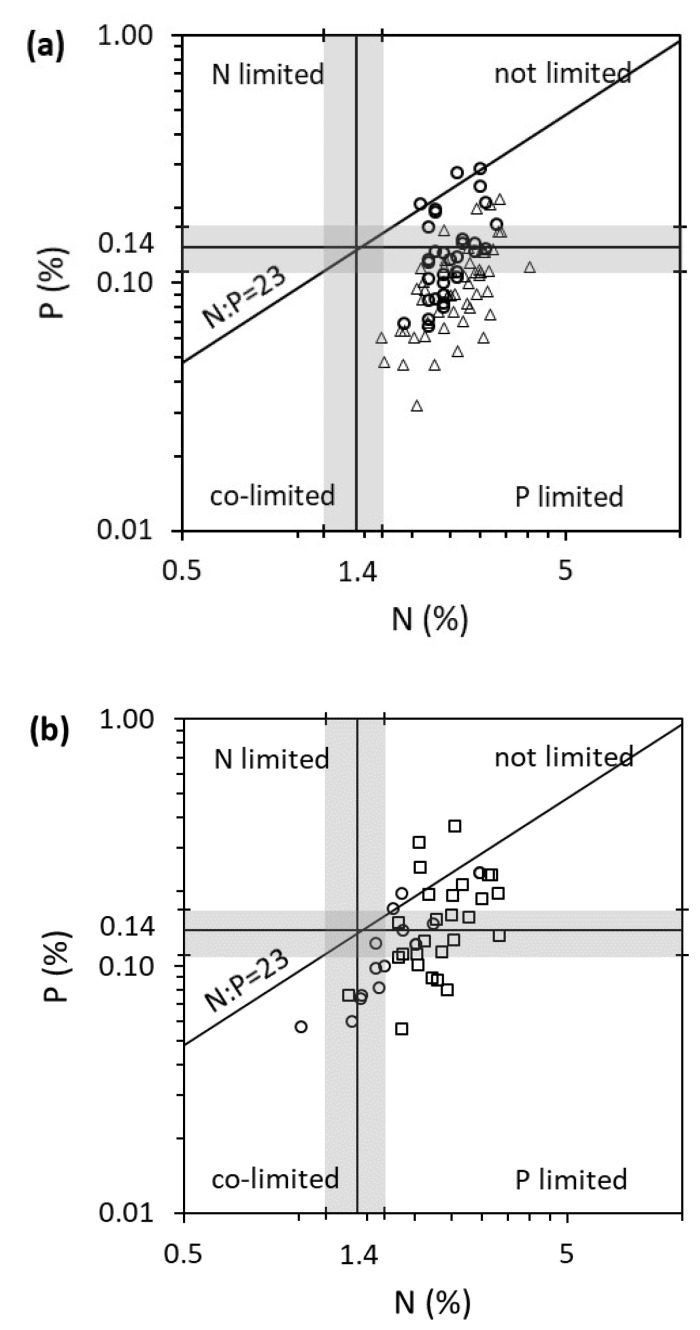
Tissue nutrient concentrations (% dry weight) from the shoots of *Juncus bulbosus* in (**a**) six lakes (*n* = 86) surveyed in spring (circles) and autumn (triangles) 2006 and (**b**) 16 lakes (circles) and 28 river sites (squares) surveyed in summer 2010. Results plotted against average (±standard error of the mean, grey shade) nutrient N (1.4 ± 0.25), P (0.14 ± 0.03) and N:P (23.4 ± 3.3) critical thresholds for maximum yield determined for hydrophytes from a literature review of laboratory assays [26].

**Figure 3 plants-10-00441-f003:**
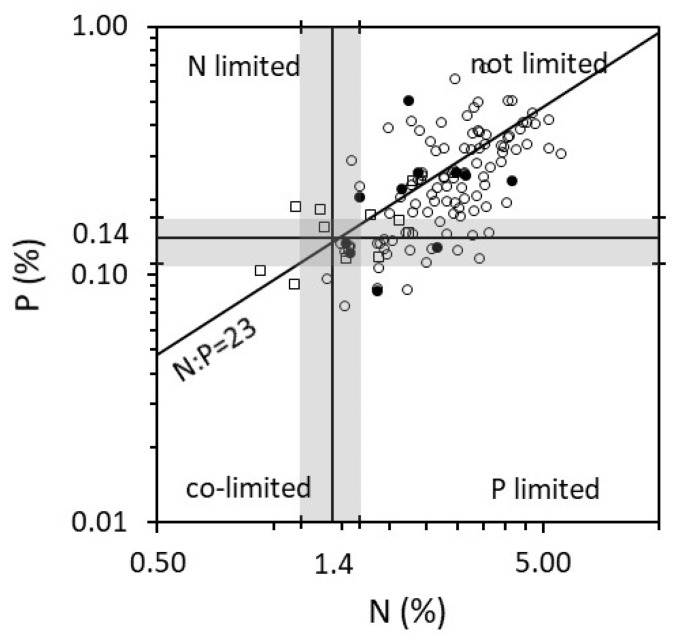
Tissue nutrient concentrations (% dry weight) from (i) the shoots of *Juncus bulbosus* (filled circles) and associated species (open circles) collected in Scotland during the summer across all types of aquatic habitats (*n* = 102 samples [14]); (ii) 12 species of hydrophytes and helophytes (open squares) from oligotrophic Spanish lakes [36]. Species list from Scotland with the number of samples (corresponding to the number of datapoints per species on the graph): *Callitriche hamulata* (8), *Isoetes lacustris* (4), *Juncus bulbosus* (11), *Littorella uniflora* (11), *Lobelia dortmanna* (5), *Myriophyllum alterniflorum* (13), *Potamogeton alpinus* (2) *P. natans* (18), *P. polygonifolius* (12), *Sparganium angustifolium* (12), *Subularia aquatica* (1), *Utricularia intermedia* (2) *Utricularia vulgaris sensu lato* (3). Same nutrient threshold as Figure 2.

**Figure 4 plants-10-00441-f004:**
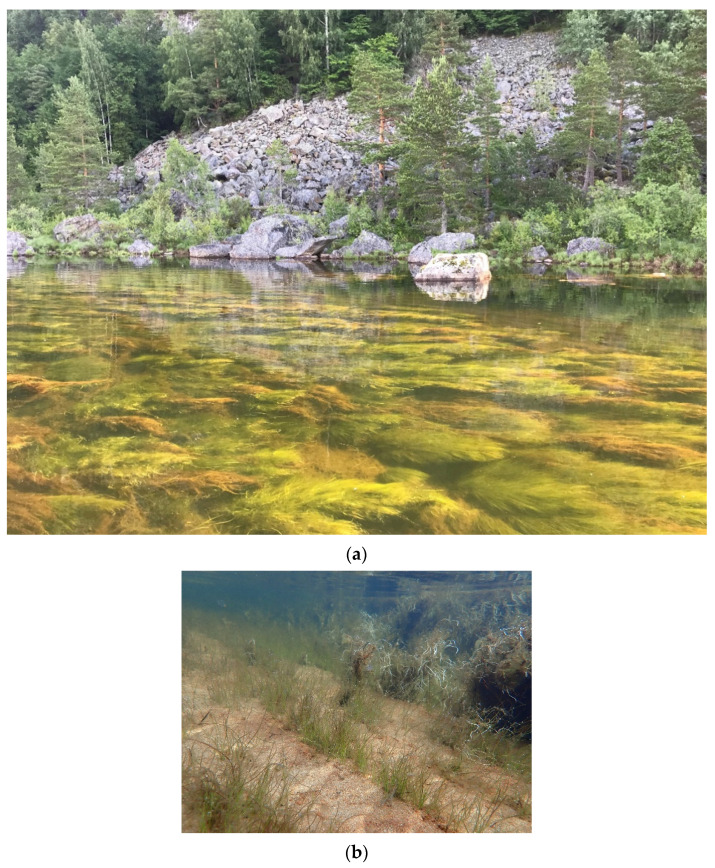
*Juncus bulbosus* in River Otra, Rysstad, Southern Norway. (**a**) Large growth form. (**b**) Rosette growth form in the foreground; also note the white roots in the background (**c**) stem with annual shoots. Photos by Benoît O.L. Demars.

**Table 1 plants-10-00441-t001:** Mixed-effects model to test the effects of the time of sampling (spring *versus* autumn) and growth forms (rosette, large growth form, new shoots) in *Juncus bulbosus*. Data from 2006 (see Methods). Bonferroni correction for multiple testing α = 0.05/12 = 0.004.

	Time of Sampling		Growth Form	
	$\chi_{1}^{2}$	*P*	$\chi_{1}^{2}$	*P*
C	0.2	0.68	1.8	1.41
N	2.3	0.13	9.9	0.007
P	20.1	7 × 10^−6^	1.5	0.47
C:N	1.9	0.17	8.1	0.02
C:P	20.2	7 × 10^−6^	1.2	0.55
N:P	33.1	9 × 10^−9^	7.2	0.03

**Table 2 plants-10-00441-t002:** Mixed-effects model to test the effects of the time of sampling (early *versus* late summer) and growth forms (rosette, large growth form, new shoots) in *Juncus bulbosus*. Data from 2008 (see Methods). Bonferroni correction for multiple testing α = 0.05/12 = 0.004.

	Time of Sampling		Growth Form	
	$\chi_{1}^{2}$	*P*	$\chi_{1}^{2}$	*P*
C	0.03	0.87	2.1	0.15
N	1.1	0.29	0.7	0.41
P	0.1	0.77	0.7	0.39
C:N	1.1	0.30	1.7	0.20
C:P	0.1	0.78	0.4	0.51
N:P	0.4	0.51	1.3	0.25

**Table 3 plants-10-00441-t003:** Percentage of variance in plant nutrient concentrations (C, N, P) and stoichiometry (C:N:P) explained by organs (root *versus* shoot), habitats (lake *versus* river) and environmental variables (water and sediment chemistry) using redundancy analyses: singly and after stepwise selection. Only significant explanatory variables are shown. *n* = 80, missing data were replaced by averages across all samples. TP = total P, EC = electric conductivity.

	Percentage of Explained Variance				
	Singly	*P*	After Stepwise Regression	*P*	Selection Order
Organs	20.8	<0.001	20.8	<0.001	1
Habitats	4.6	0.004	4.1	0.028	3
TP	4.6	0.006	3.9	0.040	4
lnEC	3.0	0.027	4.3	0.019	2

**Table 4 plants-10-00441-t004:** *Juncus bulbosus* C, N, P tissue concentrations from southern Norway. *n* = number of samples.

Datasets	Year	Description
6 lakes	2006	Replicate *J. bulbosus* sampling of different growth forms in 6 lakes at two seasons (spring and autumn), *n* = 86
10 lakes	2008	Single *J. bulbosus* samples (roots and shoots together), two sampling times (early and late summer) and two growth forms, *n* = 40
16 lakes and 28 river sites	2010	Single *J. bulbosus* samples, two habitats (lakes and rivers) and two organs (roots *versus* shoots), *n* = 80

## Data Availability

All data supporting reported results can be found in the Appendix A.

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
