# Peer review of "Juncus Bulbosus Tissue Nutrient Concentrations and Stoichiometry in Oligotrophic Ecosystems: Variability with Seasons, Growth Forms, Organs and Habitats"

_plants, 2021, doi:10.3390/plants10030441_

Round 1
Reviewer 1 Report
General
Juncus bulbosus is not typical aquatic plant, but rather amphiphyte. It is a very variable species, ranging from tufted, terrestrial plants to submerged, floating aquatics, often rooting at the nodes. It occurs in aquatic an terrestrial habitats, seasonally wet habitats, and from acidic to neutral soils. It also grows in some calcareous turloughs. Here it seems that only aquatic form f. aquatica is indicated, so this should be mentioned throughout the text.
- Add clear hypotheses
- Please explain what is the main reason for different growth forms of J. bulbosus in water in your case?
- Phosphorus limitation may also be more pronounced in lakes than rivers due to higher organic matter content in lake than river sediments (18% versus 2%, respectively) potentially triggering root iron plaque formation preventing P uptake, as for other isoetids… I think that in rivers the problem may be permanent input by runoff and transfer along river course
- Extent the description of species. What was the water depth of sampled plants? Fig. 3. What is the point of species and their frequency presented in the caption to Fig.3?
Author Response
Ref #1
General
Thank you for raising interesting comments helping to improve the manuscript
Juncus bulbosus is not typical aquatic plant, but rather amphiphyte. It is a very variable species, ranging from tufted, terrestrial plants to submerged, floating aquatics, often rooting at the nodes. It occurs in aquatic an terrestrial habitats, seasonally wet habitats, and from acidic to neutral soils. It also grows in some calcareous turloughs. Here it seems that only aquatic form f. aquatica is indicated, so this should be mentioned throughout the text.
> We agree with the description and updated section 4.1 Juncus bulbosus as follows: “J. bulbosus is an amphibious plant morphologically very plastic, ranging from small tufted terrestrial plants to submerged, floating aquatics, often rooting at the nodes and freely fruiting [60]. In this study, all samples were rooted underwater, and thus J. bulbosus may be referred to as f. aquatica, with the largest shoots reaching the water surface.”
- Add clear hypotheses
We set up clear aims at the end of the introduction which corresponded to the data analyses presented in results and interpretation. We could have added more specific hypotheses for some tests, e.g. plant tissue N in roots was likely to be lower than in above ground shoot (reason given in discussion), but for most tests it is a big ask because the results partly depend on the context which we did not know ‘a priori’ (e.g. nutrient availability, plant phenology, growth rate), e.g. should large shoot may result from fast growth rate (which likely need more energy (P rich ATP) and carbon fixation (N rich rubisco) per gram of plant mass) or slow growth rate over several years, in which case there may be no differences in plant stoichiometry between small and large shoot. Our study strived to extract as much information as possible from field data and infer processes from plant stoichiometry in the context of laboratory bioassays. Thus, we would rather keep our wording:
“Here we set out to (i) test whether Juncus bulbosus C, N, P and C:N:P in lakes can be related to growth forms (abundance) and time of year; (ii) test for differences in Juncus bulbosus elemental concentrations and stoichiometry between habitats (lakes versus rivers), plant organs (roots versus shoots) and habitats × plant organs; (iii) assess the likelihood of nutrient limitation for yield (standing biomass) using a comparative approach [29].”
- Please explain what is the main reason for different growth forms of J. bulbosus in water in your case?
Good point, we clarified that compared to previous studies insisting on the role of CO2 and NH4 for the massive expansion of Juncus bulbosus. We highlighted that phosphorus should not be taken out of the equation as understood from ecological stoichiometry principles, and this independently of the ecosystem (river or lake). It is also clear that water level and water discharge regulation in rivers impacted by hydropower have played an important role. In lakes, mass expansion of juncus bulbosus appeared in the years after liming, but after a decade or more the effect was not so conspicuous, likely because most of the organic matter had been mineralised. We tried to capture this in the discussion.
“Reading through the original publications [27, 42, 43], no factorial experiments were ever carried out to test the independent and interactive effects of CO2, N, and P on Juncus bulbosus growth rate and yield. Liming of lakes in Norway promoted J. bulbosus mass development through the mineralisation of the organic matter with production of CO2, NH4 and inorganic P in sediment pore water [27], until the organic matter became less reactive over time [44]. In the presence of sufficient N and P, CO2 can be a limiting factor [42, 43, 45]. Excess of CO2 availability relative to N and P is expected to increase the C:N and C:P of autotrophs in nutrient poor freshwater ecosystems [46, 47, 48]. General N and/or P limitations have been reported for phytoplankton of Norwegian lakes [49, 50], and inferred through J. bulbosus plant tissue nutrient concentration and stoichiometry [29].
The repeated story that mass development of J. bulbosus would only occur under high CO2 and sediment pore water NH4 concentrations [27, 30, 44, 51] can only make sense if the plant can access enough P for its growth because aquatic plant tissue C:N:P stoichiometry is rather homeostatic [14, 29], and this whatever the ecosystem, based on principles of ecological stoichiometry. Thus, in Norwegian lakes and rivers, it is likely that CO2, N and/or P could limit or co-limit the growth rate of J. bulbosus. Certainly the role of P should not be disregarded (this study, [29]). J. bulbosus has also been able to produce stands with large biomass in (unlimed) oligotrophic ecosystems through perennial growth where physical conditions allow (high light, low water turbulence) – see Figure 4, [52]. There were no significant differences in water column and sediment pore water nutrient concentrations between stands with low and high J. bulbosus biomass in rivers [53], unlike for lakes shortly after liming [27].”
- Phosphorus limitation may also be more pronounced in lakes than rivers due to higher organic matter content in lake than river sediments (18% versus 2%, respectively) potentially triggering root iron plaque formation preventing P uptake, as for other isoetids… I think that in rivers the problem may be permanent input by runoff and transfer along river course
> We agree, as written in the preceding sentence: “This may reflect the higher supply rate of nutrients in rivers compared to lakes, especially the more limiting factor P, with most samples below the critical N:P ratio – Figure 2b.”
- Extent the description of species. What was the water depth of sampled plants? Fig. 3. What is the point of species and their frequency presented in the caption to Fig.3?
> unfortunately we did not record the depth at which individual samples were collected, but mostly around 0.5 to 1 m depth
In fig. 3 we simply give an indication as to the number of datapoints represented by a species, we added the following text in the caption of Fig. 3 to be more specific: “(corresponding to the number of datapoints per species on the graph)”

Reviewer 2 Report
The manuscript “Juncus bulbosus tissue nutrient concentrations and stoichiome-2 try in oligotrophic ecosystems: variability with seasons, growth 3 forms, organs and habitats” presents an interesting problem and a large dataset of tissue nutrient stochiometry of an abundant macrophyte. However, I believe the analysis of those data is insufficient as presented. The aim of the authors was to give a more comprehensive picture than previous data, but they still present a picture from fractured investigations. A multivariate analysis over time, space, environmental conditions and plant traits should be included, otherwise there is nothing new in the manuscript than has been presented before. Also, the discussion is very superficial. How come that the implications of the nutrient stochiometry is not sufficiently explained? My biggest wondering is about other factors affecting tissue nutrient concentrations: it is not nutrient abundance but rather availability that determines, how many nutrients can be incorporated in plant tissues. Do the authors have any idea of the availability, or can they infer anything from environmental and climatic data? At least a much more thorough discussion of lakes vs. streams is necessary. Other specific suggestions are mentioned below.
Introduction:
- 36: your information in brackets is not quite clear
- 40-45: This information is somewhat detached from the rest of the text – please clarify what the point is of this example
- 52-53: what is meant by “sequence of analyses”? it would be better if you could state some of the specific patterns. It seems you criticise previous investigations but it is not exactly clear for what, and how your study is distinguished (and better) than those previous studies. It seems that your investigation is similarly fragmented (see Table 3)? Please clarify that.
It would also be great if you could explain why your sole focus was on Juncus and not typical isoetids, which you mention also. Does that limit the relevance of your study to Norway, or can your findings be transferred elsewhere, even if Juncus seems to be an exceptional plant in oligotrophic systems? Please address the significance of your study in a broader ecological context.
Results:
- 74: why not shorten the C:P ratio further to 666:477?
- 76: here you show the results for CNP, but for the year 2006 just above (line 70) they are missing. Please add them.
- 80: your presentation of P-values is rather unusual, normally significant levels are indicated by P<0.05 or P<0.001. Also, significant factors could be indicated by asterisks or in bold to increase the visual understanding of the tests
As mentioned above, I believe the data have to be presented in a more comprehensive, multivariate way. Perhaps consider collaboration with a statistician helping with the analysis. I believe much more information can be gained from your large dataset.
Discussion:
- 115-116: but your analyses show significant differences in 2006 but not in 2008 (tables 1 and 2), so I don’t understand how you can conclude that there are no differences? Or am I misunderstanding your data presentation?
- 129: please explain why this indicates slow growth
- 156 and following: I guess including such a literature survey as part of the results ought to be mentioned in M&M and results already
M&M:
- 182-187: In my opinion, this information is more needed in the introduction than here.
- 197-203: why were no plants collected at the peak of the growing season, in July/August, when biomass production is highest? What exactly is the aim of this preliminary study?
- 209: please rephrase “a single shoots”. I suppose you mean a tuft, including roots?
- 213-218, 224: I don’t know if referring for methods to other studies is accepted in the journal, but I personally find it undesirable. At least state the overall method of analysis and the number of replicates per site.
- 218: Although you show the GPS data of your sampled lakes and rivers, it would be great to also show a map indicating the sampled places, just to visualize the extend of your survey.
Author Response
The manuscript “Juncus bulbosus tissue nutrient concentrations and stoichiome-2 try in oligotrophic ecosystems: variability with seasons, growth 3 forms, organs and habitats” presents an interesting problem and a large dataset of tissue nutrient stochiometry of an abundant macrophyte. However, I believe the analysis of those data is insufficient as presented. The aim of the authors was to give a more comprehensive picture than previous data, but they still present a picture from fractured investigations. A multivariate analysis over time, space, environmental conditions and plant traits should be included, otherwise there is nothing new in the manuscript than has been presented before. Also, the discussion is very superficial. How come that the implications of the nutrient stochiometry is not sufficiently explained?
We are grateful for the numerous comments raised to improve our manuscript.
We presented a more robust approach than Schneider et al (2013), less prone to finding significant results by chance, and not biased by plant tissue analysed (entire plants versus aboveground shoot) – something rather fundamental considering that the best predictor of plant tissue stoichiometry in our study was organs (root versus shoot).
We focused on explaining the variability in plant tissue nutrient concentrations and stoichiometry, rather than trying to explain plant abundance (corresponding to growth form) as in Schneider et al (2013). The three datasets we have presented allowed for more rigorous testing of the factors listed in the title of our study, some not previously tested. We have added some multivariate analyses also including water and sediment for the 2010 dataset.
Data interpretation was based on principles of ecological stoichiometry and a literature review of laboratory bioassays recently published and applied to different data (lakes surveyed in 2007, see Moe et al 2019). We expanded the discussion on this matter to better highlight the novelty of our findings.
All the data analyses, figures, tables and interpretations produced in this study are new.
My biggest wondering is about other factors affecting tissue nutrient concentrations: it is not nutrient abundance but rather availability that determines, how many nutrients can be incorporated in plant tissues. Do the authors have any idea of the availability, or can they infer anything from environmental and climatic data? At least a much more thorough discussion of lakes vs. streams is necessary. Other specific suggestions are mentioned below.
Plant growth results from the balance of nutrient demand to nutrient availability. Limitations in plant nutrients is best determined from bioassays. We interpreted observed plant tissue nutrient concentration and stoichiometry against such bioassays (literature review published in Moe et al 2019). This is the best we could do to interpret observational field data.
We also strived to find direct links between plant tissue concentration and stoichiometry and external nutrient concentrations using the available data and multivariate analyses but for this approach to work well you need very large and uncorrelated gradients in nutrients. This was not part of the design of the present study focusing on the following predictors: variability with seasons, growth forms, organs and habitats.
We did not expand the discussion on habitats (river versus lake), but we increased the discussion on plant ecological stoichiometry with principles applicable to any ecosystem including lakes and rivers, as follows:
“Reading through the original publications [27, 42, 43], no factorial experiments were ever carried out to test the independent and interactive effects of CO2, N, and P on Juncus bulbosus growth rate and yield. Liming of lakes in Norway promoted J. bulbosus mass development through the mineralisation of the organic matter with production of CO2, NH4 and inorganic P in sediment pore water [27], until the organic matter became less reactive over time [44]. In the presence of sufficient N and P, CO2 can be a limiting factor [42, 43, 45]. Excess of CO2 availability relative to N and P is expected to increase the C:N and C:P of autotrophs in nutrient poor freshwater ecosystems [46, 47, 48]. General N and/or P limitations have been reported for phytoplankton of Norwegian lakes [49, 50], and inferred through J. bulbosus plant tissue nutrient concentration and stoichiometry [29].
The repeated story that mass development of J. bulbosus would only occur under high CO2 and sediment pore water NH4 concentrations [27, 30, 44, 51] can only make sense if the plant can access enough P for its growth because aquatic plant tissue C:N:P stoichiometry is rather homeostatic [14, 29], and this whatever the ecosystem, based on principles of ecological stoichiometry. Thus, in Norwegian lakes and rivers, it is likely that CO2, N and/or P could limit or co-limit the growth rate of J. bulbosus. Certainly the role of P should not be disregarded (this study, [29]). J. bulbosus has also been able to produce stands with large biomass in (unlimed) oligotrophic ecosystems through perennial growth where physical conditions allow (high light, low water turbulence) – see Figure 4, [52]. There were no significant differences in water column and sediment pore water nutrient concentrations between stands with low and high J. bulbosus biomass in rivers [53], unlike for lakes shortly after liming [27].”
Introduction:
- 36: your information in brackets is not quite clear
> We changed the text to “Slightly higher retention rates by aquatic plants (up to 10-13% of the dissolved inorganic N river flux during the summer) were reported in a lowland river in The Netherland [13].”
- 40-45: This information is somewhat detached from the rest of the text – please clarify what the point is of this example
> We improved the linkage: “Aquatic plant nutrient retention is often determined for the shoot without the roots, but in nutrient poor aquatic ecosystems aquatic plants tend to be dominated by isoetids characterised by high root:shoot ratios [17].”
- 52-53: what is meant by “sequence of analyses”? it would be better if you could state some of the specific patterns. It seems you criticise previous investigations but it is not exactly clear for what, and how your study is distinguished (and better) than those previous studies. It seems that your investigation is similarly fragmented (see Table 3)? Please clarify that.
> We re-phrased to “Some patterns in tissue C, N, P concentrations and C:N:P stoichiometry of J. bulbosus in Norwegian lakes and rivers were identified [30], but using averaged plant data from different years and analysed in different ways (entire plant versus shoot).”
Splitting the data as previously done increased the probability to find some significant results by chance, especially as we now know from our results that growth form (highly related to abundance) does not (or very poorly) explained C:N:P stoichiometry in the collected data.
Gathering together the 2008 and 2010 data as previously done to detect temporal changes within growth forms using root+shoot 2008 data versus only shoot 2010 data was also problematic because root and shoot have very different stoichiometry as shown here. So we analysed these data in different ways and separately.
We added some text in method: “Two of the datasets (2008, 2010) were partially used in a previous study designed to test different hypotheses, as explained in introduction [30]. Here we did not combine those datasets as in [30] as they were not comparable and we did not use plant nutrient concentrations and stoichiometry as factors to explain plant abundance (corresponding to growth form) as in [30], but as response variables to test for the effects of seasons, growth forms, organs and habitats. We then interpret those datasets differently to [30] with a comparative approach based on a review of laboratory bioassays, as previously applied to a dataset collected in 2007 [29].”
It would also be great if you could explain why your sole focus was on Juncus and not typical isoetids, which you mention also. Does that limit the relevance of your study to Norway, or can your findings be transferred elsewhere, even if Juncus seems to be an exceptional plant in oligotrophic systems? Please address the significance of your study in a broader ecological context.
Good point, we added:
“In north-west Europe few aquatic plant species grow to form large submerged stands (1-3 m tall) in oligotrophic ecosystems, and among those species Juncus bulbosus is the only (facultative) perennial, and also the only angiosperm able to colonise acid mine drainage [22], playing an important role in ecosystem structure and functions [23, 24, 25]. Since J. bulbosus is also widely distributed in northern Europe [26], it makes it an important species to study.”
Results:
- 74: why not shorten the C:P ratio further to 666:477?
> sorry we do not understand the suggestion
- 76: here you show the results for CNP, but for the year 2006 just above (line 70) they are missing. Please add them.
> Text added “C, N, P (46±2, 1.7±0.2, 0.16±0.08%) and C:N:P (1073:31:1) …”
- 80: your presentation of P-values is rather unusual, normally significant levels are indicated by P<0.05 or P<0.001. Also, significant factors could be indicated by asterisks or in bold to increase the visual understanding of the tests
> In the previous study (Schneider et al 2013) did not adjust the level of significance for multiple testing “Because each analysis represented a separate hypothesis, there was no need to adjust Ë› for multiple testing (Perneger, 1998).” Schneider et al (2013) carried out 90 tests (45 tests in Table 2 for rivers and 45 tests in Table 3 for lakes), but did not present as many hypotheses, thus some significant results (at P<0.05) may simply appear by chance.
In our study we used a somewhat conservative approach to correct for multiple testing, not least because C, N, and P concentrations and C:N, C:P, N:P are somewhat related, thus we now report the marginal and conditional R2 of the mixed effects models, including for the effect of growth form on N plant tissue which had a probability of P=0.007 (not strictly significant after correction P<0.004, but should not be entirely dismissed).
New added text in results: “The effect of growth form on N plant tissue (close to significance) explained 9.3% of the variance while the full mixed effects model (including random effects) explained 34% of the variance. This was mostly due to differences (median) in N tissue concentrations in large growth form (June: 2.2%, October: 2.6%) versus rosette (June: 2.6%, October: 2.8%). The effect of time of sampling (June versus October) on P plant tissue explained 14% of the variance and the full model explained 49% of the variance.”
New added text in method: “The fit of the models with significant fixed effects was assessed with the conditional and marginal coefficient of determination (R2) using the R function r.squaredGLMM from the MuMIn package [64, 65]. The conditional R2 represents the variance explained by fixed and random factors together; and the marginal R2 represents the variance explained solely by the fixed effects.”
As mentioned above, I believe the data have to be presented in a more comprehensive, multivariate way. Perhaps consider collaboration with a statistician helping with the analysis. I believe much more information can be gained from your large dataset.
> we have added some multivariate analyses (new Table 3) with additional text in results: “Multivariate analyses identified organs (shoot versus root) as the main factor able to explain the variance in plant tissue concentrations and stoichiometry, together with habitats (lake versus river), total P and electric conductivity – independently of the way missing data were handled – see Table 3. Forward selection of these significant factors produced a highly significant (P<0.001) model with adjR2 of 30% – see Table 3.”
The reason we had not included those analyses in this paper was that it does not show much more. Schneider et al (2013) had ruled out multivariate analyses due to issue with missing data. However, handling the few missing data in two different ways gave the same results. The authors of the present study have very strong statistical background and much novelty in our results stem from handling the data in different ways and testing for different and additional hypotheses.
Discussion:
- 115-116: but your analyses show significant differences in 2006 but not in 2008 (tables 1 and 2), so I don’t understand how you can conclude that there are no differences? Or am I misunderstanding your data presentation?
> We have explained above that we corrected for multiple testing, as indicated in the Table caption and method section. To be significant results had to have P<0.004 (corresponding to P=0.05 without correction for multiple testing).
- 129: please explain why this indicates slow growth
> Added some references and text changed to “…, likely indicative of a period with slow growth as indicated by the nutrient thresholds derived from laboratory bioassays (Figure 2a, [18, 29]).”
- 156 and following: I guess including such a literature survey as part of the results ought to be mentioned in M&M and results already
> Figure 3 was simply to put the results into the broader context of other aquatic plants generally growing in the same lakes and rivers as Juncus bulbosus, we did not use it to test specific hypotheses but to infer nutrient limitation from plant tissue nutrient thresholds and thus it should not be part of results but discussion.
M&M:
- 182-187: In my opinion, this information is more needed in the introduction than here.
> we added some additional text in introduction (see above) regarding the ecological role of Juncus bulbosus, but kept this text here and expanded it as suggested by ref#1
- 197-203: why were no plants collected at the peak of the growing season, in July/August, when biomass production is highest? What exactly is the aim of this preliminary study?
This was just a preliminary investigation, but it allowed to test for differences between seasons, unlike in Schneider et al 2013 which used data from only summer time
- 209: please rephrase “a single shoots”. I suppose you mean a tuft, including roots?
> Yes this was confusing, we meant and change the text to “A single individual” (roots and shoot)
- 213-218, 224: I don’t know if referring for methods to other studies is accepted in the journal, but I personally find it undesirable. At least state the overall method of analysis and the number of replicates per site.
> in this case, we used the same datasets (2008, 2010) as in a previous study where the methods were described in great details (Schneider et al 2013), so we did not think it was necessary to copy paste the same text into this study. Replication and total number of samples were given in Table 4. We also provided the raw data in supplementary information.
- 218: Although you show the GPS data of your sampled lakes and rivers, it would be great to also show a map indicating the sampled places, just to visualize the extend of your survey.
> the map was already published in Schneider et al 2013 for the two main datasets (2008, 2010)
Round 2
Reviewer 2 Report
The authors have taken all my comments into consideration. I have merely minor comments.
Abstract line 18: Biomass instead of mass
Introduction line 50: Mentioning acid mine drainage here is very confusing and out of context, I would erase it. Otherwise it must be put into context.
Results line 78: it seems something is missing here, do you mean "N in plant tissue"? Check the language.
In my opinion, this manuscript can be accepted now.
Author Response
Thank you for careful reading
Abstract line 18: Biomass instead of mass
reply: by "mass development" we mean very large development, so we left it as is
Introduction line 50: Mentioning acid mine drainage here is very confusing and out of context, I would erase it. Otherwise it must be put into context.
reply: OK, we deleted it and associated references
Results line 78: it seems something is missing here, do you mean "N in plant tissue"? Check the language.
reply: yes, we rephrased "N concentrations in plant tissue" and two lines below "P concentrations in plant tissue"